# Subtilisin of *Leishmania amazonensis* as Potential Druggable Target: Subcellular Localization, In Vitro Leishmanicidal Activity and Molecular Docking of PF-429242, a Subtilisin Inhibitor

Pollyanna Stephanie Gomes [1,2,3,4,†], Monique Pacheco Duarte Carneiro [1,4,5,†], Patrícia de Almeida Machado [1,2,4], Valter Viana de Andrade-Neto [6], Alessandra Marcia da Fonseca-Martins [1,2,3,4], Amy Goundry [5], João Vitor Marques Pereira da Silva [7], Daniel Claudio Oliveira Gomes [8], Ana Paula Cabral de Araujo Lima [5], Vítor Ennes-Vidal [9], Ana Carolina Rennó Sodero [7], Salvatore Giovanni De-Simone [10,11,12] and Herbert L. de Matos Guedes [1,2,3,4,*]

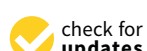



1 Laboratório de Imunologia Clínica, Instituto Oswaldo Cruz, Fundação Oswaldo Cruz—Fiocruz, Rio de Janeiro 21040-360, Brazil; pollyannaufrj@gmail.com (P.S.G.); moniquepdc@gmail.com (M.P.D.C.); patriciamachado12@yahoo.com.br (P.d.A.M.); alemfmartins@gmail.com (A.M.d.F.-M.)
2 Laboratório Interdisciplinar de Pesquisas Médicas, Instituto Oswaldo Cruz, Fundação Oswaldo Cruz, Rio de Janeiro 21040-360, Brazil
3 Laboratório de Imunofarmacologia, Instituto de Biofísica Carlos Chagas Filho IBCCF, Universidade Federal do Rio de Janeiro, Rio de Janeiro 21941-170, Brazil
4 Laboratório de Imunobiotecnologia, Instituto de Microbiologia Paulo de Goés, Universidade Federal do Rio de Janeiro, Rio de Janeiro 21941-902, Brazil
5 Laboratório de Bioquímica e Biologia Molecular de Proteases, Instituto de Biofísica Carlos Chagas Filho IBCCF, Universidade Federal do Rio de Janeiro, Rio de Janeiro 21941-170, Brazil; amy.goundry@gmail.com (A.G.); anapaula@biof.ufrj.br (A.P.C.d.A.L.)
6 Laboratório de Bioquímica de Tripanossomatídeos, Instituto Oswaldo Cruz, Fundação Oswaldo Cruz, Rio de Janeiro 21040-360, Brazil; valter@ioc.fiocruz.br
7 Faculdade de Farmácia, Universidade Federal do Rio de Janeiro, Rio de Janeiro 21941-170, Brazil; joao.v.marques98@hotmail.com (J.V.M.P.d.S.); acrsodero@gmail.com (A.C.R.S.)
8 Núcleo de Doenças Infecciosas, Universidade Federal do Espírito Santo, Vitória 29047-100, Brazil; dgomes@ndi.ufes.br
9 Laboratório de Estudos Integrados em Protozoologia, Instituto Oswaldo Cruz, Fundação Oswaldo Cruz, Rio de Janeiro 21040-360, Brazil; vitorennesvidal@gmail.com
10 Center for Technological Development in Health (CDTS), National Institute of Science and Technology for Innovation on Diseases Neglected Population (INCT-IDPN), FIOCRUZ, Rio de Janeiro 21040-900, Brazil; Salvatore.simone@fiocruz.br
11 Epidemiology and Molecular Systematic Laboratory, Oswaldo Cruz Institute, FIOCRUZ, Rio de Janeiro 21040-900, Brazil
12 Cellular and Molecular Biology Department, Biology Institute, Federal Fluminense University, Niterói 24020-141, Brazil
* Correspondence: herbert@ioc.fiocruz.br or herbert@micro.ufrj.br
† These authors contributed equally to this work.

**Abstract:** Subtilisin proteases, found in all organisms, are enzymes important in the post-translational steps of protein processing. In *Leishmania major* and *L. donovani*, this enzyme has been described as essential to their survival; however, few compounds that target subtilisin have been investigated for their potential as an antileishmanial drug. In this study, we first show, by electron microscopy and flow cytometry, that subtilisin has broad localization throughout the cytoplasm and membrane of the parasite in the promastigote form with foci in the flagellar pocket. Through in silico analysis, the similarity between subtilisin of different *Leishmania* species and that of humans were determined, and based on molecular docking, we evaluated the interaction capacity of a serine protease inhibitor against both life cycle forms of *Leishmania*. The selected inhibitor, known as PF-429242, has already been used against the dengue virus, arenaviruses, and the hepatitis C virus. Moreover, it proved to have antilipogenic activity in a mouse model and caused hypolipidemia in human cells in vitro. Here, PF-429242 significantly inhibited the growth of *L. amazonensis* promastigotes of four different

strains (IC$_{50}$ values = 3.07 ± 0.20; 0.83 ± 0.12; 2.02 ± 0.27 and 5.83 ± 1.2 μM against LTB0016, PH8, Josefa and LV78 strains) whilst having low toxicity in the host macrophages (CC$_{50}$ = 170.30 μM). We detected by flow cytometry that there is a greater expression of subtilisin in the amastigote form; however, PF-429242 had a low effect against this intracellular form with an IC50 of >100 μM for intracellular amastigotes, as well as against axenic amastigotes (94.12 ± 2.8 μM for the LV78 strain). In conclusion, even though PF-429242 does not affect the intracellular forms, this drug will serve as a tool to explore pharmacological and potentially leishmanicidal targets.

**Keywords:** *Leishmania*; serine protease; subtilisin; PF-429242; cellular localization

## 1. Introduction

*Leishmania* spp. is an obligate intracellular parasite that resides within host cells. The clinical forms of leishmaniasis can be divided basically into three different types: cutaneous (CL), mucocutaneous (MCL), and visceral (VL), the latter of which can be lethal if not treated. The most common form around the world is CL, characterized by ulcerative, painless, single, or multiple lesions [1]. Currently, 92 and 83 countries or territories are endemic for CL and VL, respectively [2]. Over 1 billion people live in leishmaniasis endemic areas, where about 95% of CL cases occur [2]. Annually, between 700,000 and 1 million new cases arise. In 2020, of all the endemic countries, Brazil was present in the ranking of the ten countries with the highest number of VL and CL cases [2]. One of the major etiologic agents of American tegumentary leishmaniasis in Brazil is *Leishmania (Leishmania) amazonensis* [3,4], although the infection caused by this species has already been described as having a wide spectrum of clinical manifestations, including the visceral form [5,6].

Leishmaniasis treatment is challenging because the available drugs can cause several side effects, such as cardiotoxicity, hepatotoxicity, and nephrotoxicity [7]. In addition, they are associated with high costs and unsatisfactory results as they have low selectivity, thus requiring higher doses, whereas the routes of administration are painful. These drawbacks mean patients often drop out of treatment [8], leading to the rise and spread of drug resistance [9]. Due to the great burden of leishmaniasis worldwide and the difficulties related to treatment, new therapeutic strategies are needed.

Parasite proteases have been demonstrated to be involved in parasite-host interaction [10]. The use of different protease inhibitors (natural or synthetic) has been reported to cause a significant reduction in the virulence of pathogens by modifying the pathophysiology of diseases, indicating their potential as potent and efficacious antiparasitic drugs [11]. Serine proteases, in particular, are important virulence factors in parasitic diseases. In *Plasmodium*, the merozoite surface protein-1 (MSP-1) is crucial for the merozoite egress: in order to achieve its function, it must first be cleaved by a parasite-derived serine protease [12]. *Toxoplasma* serine proteases also appear to mediate the release of secretory proteins from the rhoptries, which is important in the penetration process [13]. Viral infections also rely on serine proteases, such as NS3 of the dengue virus [14], whereas the coronaviruses that cause SARS and MERS depend on serine proteases located at the host cell surface for their activation [15].

*Leishmania* spp. have between 26 and 28 serine protease genes [16]. Through comparative genomic analysis of four species of *Leishmania*, Silva-Almeida and coworkers (2014) observed 15 conserved alleles of serine proteases, which were predominantly grouped on chromosome 28 [17]. Serine proteases have been classified into evolutionarily unrelated clans, and can be subdivided into families, based on their catalytic mechanism. The clans differ in terms of the general fold and the order of the catalytic residues in the primary sequence; a catalytic triad of serine proteases is typically composed of histidine, serine, and aspartic residues in the active site [18]. Despite this highly evolutionarily conserved triad in the active site, studies have shown unusual folds unrelated any other serine protease, with

an active site consisting instead of histidine, an aspartic acid, or even a tetrad consisting of serine, two histidine, and aspartic acid, as described in the cytomegalovirus [19,20].

In *Leishmania* spp., the serine proteases are distributed in 6 clans among 8 families, and they represent one of three protease groups within the genome, consisting of 10–16% of all the protease genes. Until now, only functions of the SB, SC, and SF clans have functions demonstrated in the genus [16].

In *Leishmania*, there are two isoforms of genes from each clan and the SB clan includes the S8 family, known as the subtilase family, which is the second largest family of serine peptidases [16]. Two genes encoding subtilisin (SUB) have been identified in *L. major* (Q4Q827 and Q4QG50); however, only Q4QG50 has the S8 domain [21]. Accordingly, Swenerton and coworkers (2010) observed that SUB is required for the normal regulation of the trypanothione reductase system of *Leishmania*, as it appears that SUB has the function of processing the terminal peroxidases [22]. In this study, they generated a SUB knockout mutant for *L. donovani* (SUB$^{-/-}$) and attempted the same in *L. major*; however, only one allele could be deleted (SUB$^{+/-}$), despite multiple attempts at targeting the second allele [22]. *L. donovani* SUB$^{-/-}$ mutants demonstrated a reduced ability to differentiate from promastigotes into amastigotes in vitro. In addition, the resulting axenic amastigotes had abnormal membranes, retained flagella, and increased binucleation [22].Thus, SUB appears to be involved in maintaining the survival and viability of *Leishmania*.

To investigate the potential of a known inhibitor of serine proteases that could target SUB and have antileishmanial activity, we chose PF-429242 dihydrochloride, which is described to be an inhibitor of membrane-bound transcription factor site-1 protease (MBTP1—human subtilisin) [23]. Moreover, PF-429242 has already been shown to be effective in the suppression of viral replication in cells infected with hepatitis C virus (HCV) [24], Lassa virus [25], lymphocytic choriomeningitis virus [26], arenaviruses [27], and dengue virus [28], and more recently we demonstrated that it significantly inhibited the growth of promastigotes and intracellular amastigotes of *L. infantum* [29].

Thus, in this study, we will evaluate the ability of PF-429242 in binding *Leishmania* SUB through molecular docking and determine the cellular location of the S8 domain-containing SUB in *L. amazonensis*. Thereafter, to determine whether SUB could be a potential druggable target, we use the serine protease, PF-429242, as an investigative pharmacological tool against the promastigotes and amastigotes of *L. amazonensis*.

## 2. Material and Methods

### 2.1. Reagents

PF-429242 dihydrochloride was purchased from Sigma-Aldrich (St. Louis, MO, USA) and dissolved in deionized water. MTT (3-(4,5-dimethylthiazol-2-yl)-2,5-diphenyltetrazolium bromide) and resazurin (7-hydroxy-3H-phenoxazin-3-one-10-oxide sodium salt) were both purchased from Sigma-Aldrich.

### 2.2. Parasites

Four strains of *L. amazonensis* were used: MHOM/BR/77/LTB0016, MHOM/BR/75/Josefa, IFLA/BR/1967/PH8 and RAT/BA/74/LV78. The parasites were obtained from the footpad lesions of infected BALB/c mice (Ethics Committee on Animal Experimentation—CEUA of UFRJ, protocol number 080/2018) and then maintained as promastigotes in culture. *L. amazonensis* Josefa strain was cultured in M199 medium (Sigma-Aldrich, St. Louis, MO, USA) supplemented with 0.02% hemin, 10% heat-inactivated fetal bovine serum (FBS) (Cultilab, Sao Paulo, Brazil), 100 U/mL penicillin and 100 µg/mL streptomycin. *L. amazonensis* LTB0016 was cultured in RPMI-1640 medium (Sigma-Aldrich, St. Louis, MO, USA) supplemented with 10% FBS, 100 U/mL penicillin, 100 µg/mL streptomycin, 5 mg/mL hemin, 0.5 mg/mL folic acid, 0.2 mg/mL D-biotin and 4 mg/mL adenine (Sigma-Aldrich, St. Louis, MO USA). *L. amazonensis* PH8 and LV78 strains were grown in Schneider's *Drosophila* medium (Thermo Fisher Scientific, Waltham, MA, USA) supplemented with 20% FBS, 100 U/mL penicillin and 100 µg/mL streptomycin. All promastigote

cultures were incubated at 26 °C and passaged to a new medium twice per week. Promastigotes were used until third passage in culture. Axenic amastigotes of *L. amazonensis* LV78 strain were cultured in Grace's Insect Media (Invitrogen, Carlsbad, CA, USA), pH 5.3, supplemented with 20% heat-inactivated FBS and 50 U/mL streptomycin, then maintained at 32 °C [30]. The amastigotes were used until the fifth passage in culture.

### 2.3. Multiple Sequence Alignment

The complete sequence of the SUB of *L. amazonensis* was analyzed for the percentage of identity and degree of similarity with other sequenced members of the serine protease family present in the SWISSPROT and Tr-EMBL protein banks. Database searches were performed using programs from the BLAST family. The multiple alignment was performed using the Clustal O (1.2.4) program in order to identify conserved regions, such as the catalytic triad and specificity subsites.

### 2.4. Cloning and Protein Expression of the Catalytic Domain from Subtilisin

The open reading frame (ORF) of SUB was amplified from the genomic DNA of *L. major* by PCR using oligonucleotides based on the N-terminal and C-terminal of the *L. major* gene sequences (LMJF_13_1040) deposited in the UnitProt protein database (Q4QG50). The catalytic domain (amino acids 88–409) alone was also amplified (forward primer with *Nde*l restriction site (underline): GGGA TCCCATATGTTCTCTGGCCGTGGCGTCCGCGTG; reverse primer with *Eco*RI restriction site (underline) and stop codon (in italic): GGAATTC*cta*GAGCGCCACGACTC CGGCCACGAT). The amplified DNA fragments (ORF, and catalytic domain 1302 bp) were purified from an agarose gel using the QIAquick Gel Extraction kit (Qiagen, Germantown, MD, USA), cloned into the pGEM-T vector (Promega, Madison, WI, USA) and transformed into *E. coli* DH5α. Plasmids were then purified from *E. coli* cultures using the Miniprep DNA Extraction kit (Qiagen, Germantown, MD, USA) and sequenced. The sequencing reactions were performed in an ABI PRISM 377 Sequencer (PE Biosystems, Foster City, CA, USA) using the BigDye Terminator Cycle Sequencing kit with AmpliTaq DNA polymerase (enzyme FS) (PE Biosystems, Foster City, CA, USA), according to the manufacturer's protocol.

The catalytic domain of SUB was released from the pGEM-T vector using the restriction enzymes *Nde*I and *Eco*RI (Promega, Madison, WI, USA) and subsequently ligated into a similarly digested pET28a vector using T4 DNA ligase (Promega, Madison, WI, USA). For protein expression, the construct was transformed into BL21(DE3) or Rosette-gami 2 strains of *E. coli*. The positive clones were induced with 1 mM IPTG (varying time and temperature), and the conditions that resulted in the highest level of expression of proteins in soluble form were selected. The recombinant N-terminal his-tagged proteins were isolated using an Ni-NTA Superflow resin affinity column (Qiagen, Germantown, MD, USA) using binding buffer (Tris-HCl pH 8.0, 250 mM NaCl and 5 mM Imidazole) and elution buffer (Tris-HCl pH 8.0, 250 mM NaCl and 200 mM Imidazole). When necessary, gel filtration chromatography steps using the Shim-pack Diol 150 column (Shimadzu, Torrance, CA, USA) were applied.

### 2.5. Anti-Subtilisin Sera

Polyclonal anti-subtilisin antibodies were generated in rabbits through four subcutaneous immunizations with the recombinant SUB catalytic domain (500 μg/dose with 7 days intervals). The first immunization was used in complete Freund's adjuvant (SC-24018, Santa Cruz Biotechnology, Dallas, TX, USA). The following immunizations were used together with incomplete Freund's adjuvant (Santa Cruz Biotechnology, Dallas, TX, USA). Six days after the last immunization, the animal was euthanized, and the serum was collected. Confirmation of the production of specific antibodies was assessed by Western blot.

### 2.6. Western Blot

*L. amazonensis* promastigote lysate (30 μg) was separated on an 8–10% SDS-PAGE gel and transferred onto a nitrocellulose membrane (BIORAD, Hercules, CA, USA). The

membrane was incubated with the primary antibody in 0.5% non-fat milk (rabbit polyclonal anti-subtilisin, 1:200) overnight at 4 °C, washed, and incubated with the anti-rabbit HRP secondary antibody (1:4000) in Tris-buffered saline with Tween-20 (TBST) for 1 h and revealed with ECL reagent (GE Healthcare, Chicago, IL, USA, RPN2106).

### 2.7. Flow Cytometry and Immunofluorescence Analysis (IFA)

*L. amazonensis* ($1 \times 10^6$ cells) parasites were processed and analyzed for flow cytometry and IFA as previously described [31]. Briefly, parasites fixed with 0.4% paraformaldehyde, permeabilized or not with 0.01% Triton X-100, were incubated at room temperature for 2 h with the anti-subtilisin polyclonal antibody (1:400). Cells were washed three times with phosphate-buffered saline (PBS), then incubated with Alexa 488-labeled goat anti-rabbit IgG secondary antibody (1:750) for 1 h at room temperature.

For flow cytometry, data acquisition and analysis were performed on a FACSCalibur flow cytometer equipped with a 15 mW argon laser emitting at 488 nm (BD Bioscience, San Jose, CA, USA). The omission of the primary antibody was used as a negative control. Each experimental population was first mapped using a two-parameter histogram of forward-angle light scatter versus side scatter. The mapped population ($n = 10,000$) was then analyzed for log green fluorescence using a single parameter histogram, and the mean fluorescence intensity (MFI) of each experimental system was divided by the MFI from the auto-fluorescence controls to obtain the variation index.

For IFA, the cells were incubated with DAPI solution at 1:20,000 (Sigma-Aldrich, St. Louis, MO, USA), adhered on poly-L-lysine-coated coverslips and mounted with ProLong® Gold Antifade (Molecular Probes, Eugene, OR, USA). Images were captured on a Zeiss AxioImager 4.8 confocal microscope (Oberkochen, Germany) in the Confocal Platform of the Instituto Oswaldo Cruz—Fiocruz.

### 2.8. Transmission Electron Microscopy (TEM)

The TEM was performed as described previously [32]. Briefly, *L. amazonensis* promastigotes were fixed with 4% paraformaldehyde, 0.1% glutaraldehyde in 0.1 M sodium cacodylate buffer, pH 7.2, dehydrated in methanol, embedded in resin (Lowicryl K4M) and cut. The sections were incubated with anti-subtilisin polyclonal antibody (1:200) in PBS for 2 h. After that, they were washed and incubated with anti-rabbit IgG antibodies bound to gold particles. The samples were observed on a Zeiss EM10C transmission electron microscope (Oberkochen, Germany).

### 2.9. Structure Prediction of Subtilisin

The *L. major* subtilisin and human subtilisin serine protease (S1P—also known as MBTP1,SKI1) amino acid sequences were retrieved from the UniProt database [33], under the codes, Q4QG50 and Q14703, respectively. The *L. amazonensis* sequence was obtained from the genome [34]. The S8 domain of the *L. major* and *L. amazonensis* SUBs were predicted by the Pfam server [35]. The ab initio modeling of the S8 domain was carried out using the fully automated Robetta server, which is available online (http://robetta.bakerlab.org) [36]. As the active site determination is crucial to study protein–ligand interactions, the DoGSiteScorer server [37] was used to predict the structural pockets and cavities of the SUB models. This tool combines pocket prediction, characterization and druggability estimation. The catalytic triad was predicted by comparison with the human model. Protein electrostatic surfaces were analyzed using the APBS software package in the PyMOL program (The PyMOL Molecular Graphics System, Version 1.7.5.0, Schrödinger, LLC, San Francisco, CA, USA) in order to map the features of each catalytic site.

### 2.10. Molecular Docking

Docking simulations of the inhibitor, PF-429242, at the binding site of the SUB were carried out using AutoDock Vina [38]. The 3D structure of the inhibitor was built using the Avogadro program [39]. Hydrogens were added, considering the pH as 7.2, and the

structure was minimized by the steepest descent method. Hydrogen atoms were also added to protein, and non-polar hydrogen atoms were merged. Protein coordinates were set to be rigid, whereas ligand bonds were set to be rotatable, and a grid box of 22.5 × 22.5 × 22.5 Å was centered on the cavity predicted by the DoGSiteScorer server. The number of generated binding modes was set to 100.

### 2.11. Cytotoxicity Assay

Peritoneal macrophages from BALB/c mice were obtained from the peritoneal cavity as described before [40], and $2 \times 10^6$ cells/mL prepared in RPMI with 10% FBS were distributed in 96-well plates. The plates were incubated at 37 °C with 5% $CO_2$ for 1 h, after which the wells were washed three times with PBS to remove non-adherent cells then incubated again in RPMI/10% FBS. The following day, the cells were washed with PBS and PF-429242 was added at different concentrations in RPMI/10% FBS (200; 100; 50; 25; 12.5; 6.25; 3.12; 1.56; and 0.78 µM). After 72 h at 37 °C, resazurin (final concentration 50 µM per well) or MTT (50 µg per well; Sigma-Aldrich, St. Louis, MO, USA) were added and the plates incubated again at 37 °C for 3 h or 4 h, respectively. MTT reaction was stopped by the addition of 0.7% isopropanol/HCl solution. The reactions were read on a SpectraMax fluorometer at excitation/emission of 560/590 nm for resazurin or on a SpectraMax spectrophotometer at 570 nm for MTT. The inhibition was calculated by comparing the percentage of cell death to the untreated control.

### 2.12. Promastigote Viability

Promastigotes ($2 \times 10^6$ cells/mL) of *L. amazonensis* (LTB0016, PH8, Josefa e LV78) from log-phase culture were incubated for 72 h at 25 °C in 96-well plates with different concentrations of PF-429242 (100–0.01 µM) prepared in RPMI-1640 medium without phenol red (Sigma-Aldrich, St. Louis, MO, USA) containing 10% FBS. Cultured promastigotes without the addition of PF-429242 were used as a control. Afterwards, MTT was added, and the plates incubated at 25 °C for 4 h. The reaction was stopped using a 0.7% isopropanol/HCl solution and the reading was carried out in a SpectraMax spectrophotometer at 570 nm. Results were calculated as the inhibition percentage of promastigote growth compared with the untreated control.

### 2.13. Anti-Amastigote Activity

Peritoneal macrophages ($2 \times 10^6$ cells/mL), obtained as described before [40], were distributed over 13 mm glass coverslips in 24-well plates and incubated at 37 °C with 5% $CO_2$. After 1 h, the cells were washed then incubated overnight before being washed again. The cells were then incubated with *L. amazonensis* promastigotes (LTB0016, Josefa and PH8) in the stationary phase of growth at a 5:1 ratio for 4 h at 33 °C and 5% $CO_2$. Afterwards, wells were washed with PBS to remove free parasites and PF-429242 was added at different concentrations (100; 10; 1; 0.1; and 0.01 µM) in RPMI/10% FBS. Infected cells in the absence of the inhibitor were used as a control. After 72 h of treatment at 33 °C and 5% $CO_2$, the coverslips were removed and the cells stained with Panotico (Laborclin, Rio de Janeiro, Brazil) according to the manufacturer's instructions. The coverslips were mounted on slides and for each coverslip the number of parasites within a total of 100 macrophages was counted. Results were calculated as the percentage of inhibition of amastigote growth compared with the untreated controls and $IC_{50}$ values.

### 2.14. Statistical Analysis

Statistical analyses were performed with GraphPad Prism 8 software (GraphPad Software, Inc., La Jolla, CA, USA) or GraFit 5 software. In order to select the most suitable tests, the normality and homogeneity of variances were first checked. Differences between mean values were evaluated by the use of a parametric Student's *t*-test (two-tailed) or one-way ANOVA. Results are expressed as mean ± one standard deviation (SD), and differences between the control and treated groups were considered statistically significant

when $p \leq 0.05$. $IC_{50}$ values were obtained by non-linear regression in the GraphPad Prism 8 program.

## 3. Results

### 3.1. Sequence Aligment of the Catalytic Domain of Subtilisin

Through a database search (Swiss-Prot), we determined the sequence similarity of the SUB catalytic domain between *L. mexicana*, *L. major*, *L. donovani*, *L. infantum*, *L. braziliensis*, *L. panamensis*, *Trypanosoma cruzi* and *T. brucei* as compared with *L.amazonensis* by BLAST (Table S1, See Supplementary Materials). Although the degree of conservation was not so high for *Trypanosoma* (*T. cruzi* and *T. brucei*), with around only 36% identity, there was high sequence conservation among the *Leishmania* species, greater than 78% (Figure S1). The sequence data described are available under the accession numbers in Table S1.

### 3.2. Detection of Subtilisin in L. amazonensis

The catalytic domain of *L. major* SUB (Q4QG50) was cloned into an expression vector (Figure S2A—lane 2) and the recombinant protein was purified by affinity chromatography on a nickel column (Figure S2B—lane 3), which was used to generate polyclonal anti-subtilisin antibodies.

In order to detect the presence of SUB in *L. amazonensis*, the anti-subtilisin antibody was used against cell lysates in Western blot analysis (Figure 1A). As expected, a band around 150 kDa was identified. In addition, lower and higher bands were detected, these are possibly aggregates or degradation products of the same protein which may be indicative of SUB cleavage in other sites. We performed Western blot with the anti-subtilisin antibody in recombinant *L.major* SUB, and catalytic domain was also recognized (Figure S2C). In order to determine whether the SUB of *L. amazonensis* was located intracellularly or at the cell surface, flow cytometry was performed using promastigotes that were permeabilized, or not, then labeled with the anti-subtilisin antibody (Figure 1B). Although there was no difference, there seems to be a slight increase intracellularly.

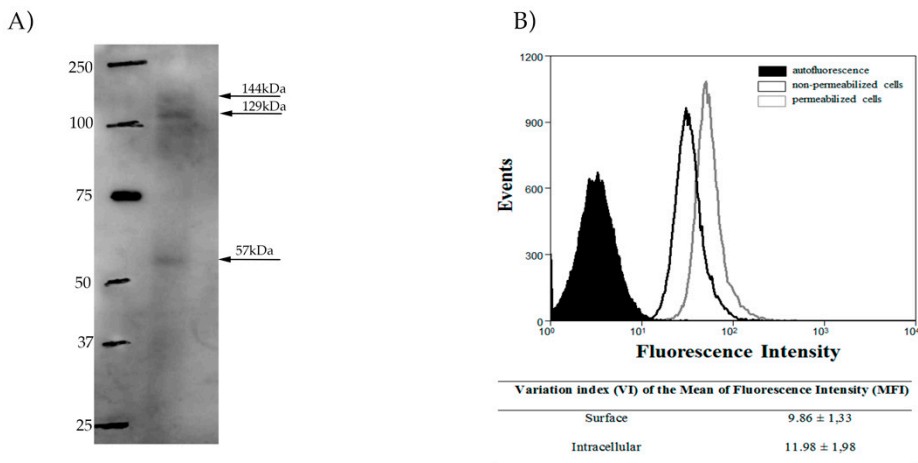

**Figure 1.** Detection of subtilisin in *L. amazonensis*. (**A**) Western blot analysis of *L. amazonensis* (MHOM/BR/77/LTB0016) promastigotes (30 μg of lysate) using an anti-subtilisin antibody (1:500) and an anti-rabbit secondary antibody (1:4000). The subtilisin-like protein bands are indicated by the black arrows. (**B**) Flow cytometry analysis of Triton X-100-permeabilized (grey line) and non-permeabilized (black line) *L. amazonensis* promastigotes labeled with the anti-subtilisin antibody (1:400) and an Alexa 488 secondary antibody (1:750). The variation index of the mean fluorescent intensity (MFI) was obtained by the division of the MFI from labeled parasites by the non-stained autofluorescence control (black fill). Experiments were performed in triplicate and graphs are representative of three independent experiments.

*3.3. Sub-Cellular Localization of Subtilisin in the L. amazonensis*

Through immunofluorescence, it was seen that SUB of *L. amazonensis* was expressed throughout the parasite body, especially an accumulation around in the flagellar pocket (Figure 2A,B).

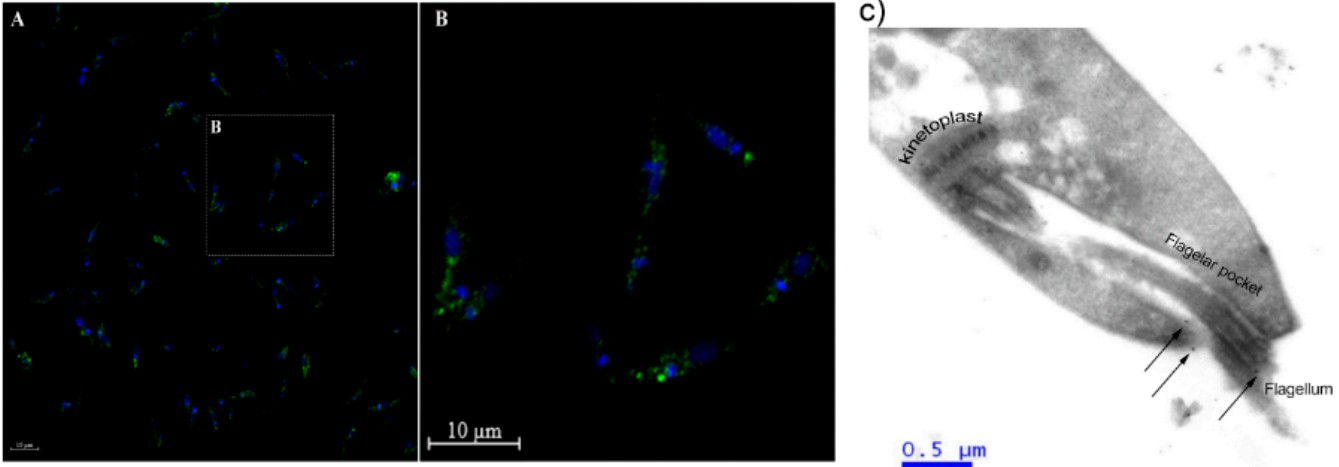

**Figure 2.** Immunolocalization of *L. amazonensis* subtilisin. (**A**) Paraformaldehyde-fixed *L. amazonensis* promastigotes were permeabilized with Triton X-100 and labeled with anti-subtilisin antibody (1:400) and an Alexa 488 secondary antibody (1:750). The parasites were then analyzed by immunofluorescence microscopy. Parasites treated with the secondary antibody alone presented no fluorescence intensity (data not shown). The images shown are representative of three independent experiments. Statistical analysis by Student's *t*-test $p < 0.05$. (**B**) The labeling of the cell cytoplasm, in detail. Bar represents 10 μm (**C**) Transmission electron microscopy of *L. amazonensis* promastigotes using anti-subtilisin (1:200) and gold-conjugated anti-IgG antibodies (arrows). Scale 0.5 μm.

In order to confirm this concentration close to the flagellar pocket, immunolocalization by TEM was performed (Figure 2C). The greatest concentration of SUB was indeed found in the flagellar pocket (black arrows). The same was also observed with confocal microscopy to detect the SUB of *L. major* (Figure S3).

*3.4. Molecular Modeling*

PF-429242 has been shown to inhibit human subtilisin serine protease, S1P [41,42]. Here, molecular modeling studies were carried out to evaluate the binding mode of PF-429242 in the SUBs of *L. major* and *L. amazonensis*. The S8 domain, which contains the catalytic triad essential for serine endopeptidase activity, was identified through the Pfam server. In the S8 domain, human S1P has 259 amino acid residues, whereas *L. amazonensis* has 371 and *L. major* has 381. The 3D structures of the S8 domain of the SUB were obtained by the Robetta server. The selected models showed good stereochemical evaluation by Ramachandran plot (more than 92% of residues located in favored regions) and an adequate position of the catalytic site. It is important to highlight that the pH value of 7.2 had to be considered, since the protonation of the catalytic histidine is located on the delta nitrogen. This is the most favorable scenario for the catalysis by serine protease, in which the imidazole ring of histidine is stabilized by the aspartate of the catalytic triad.

The three SUB models (Figure 3A–C) showed 9 β-sheets. Additionally, 8, 14, and 13 α-helix structures were identified in the human, *L. amazonensis* and *L. major* models, respectively. No disulfide bonds were identified in any of the cysteine residues of the sequences.

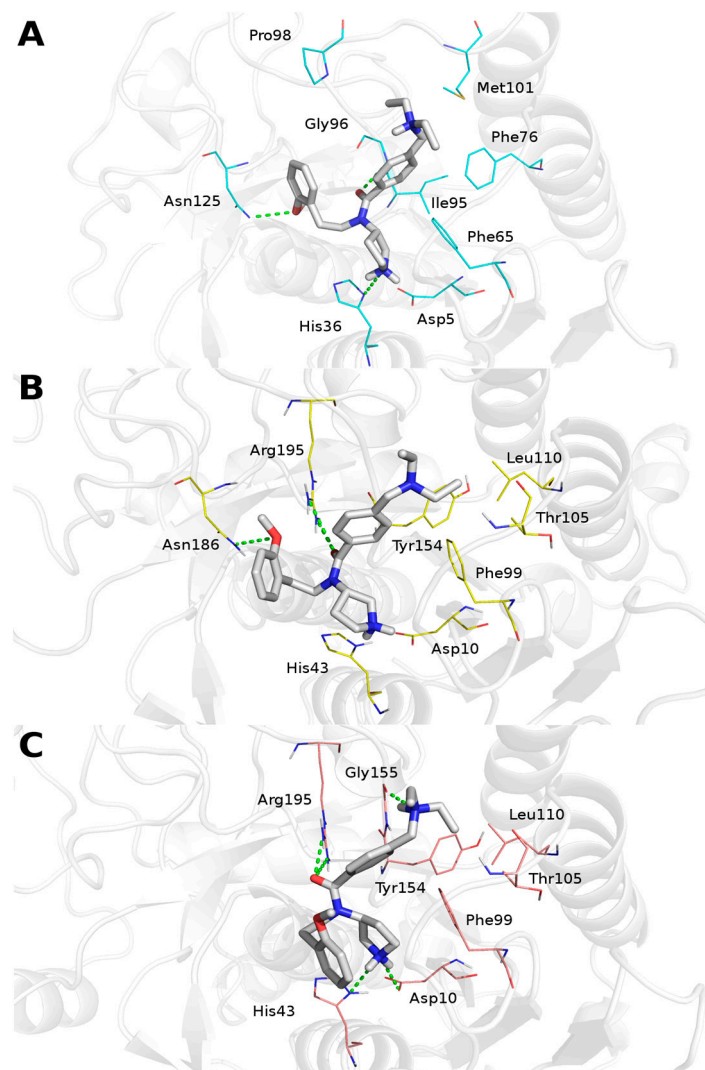

**Figure 3.** Suggested binding mode of PF-429242 (carbons in gray) in the subtilisin binding site. (**A**) Human subtilisin S1P (carbons in cyan), (**B**) *L. amazonensis* SUB (carbons in yellow), and (**C**) *L. major* SUB (carbons in pink). Protein residues in close contact with the inhibitor are shown as lines and labeled. Hydrogen bonding interactions are shown as a dotted green line. Secondary structures are shown as a transparent cartoon.

Residues involved in the active binding site were indicated by the DogSiteScorer server (Table 1). The site in the human SUB showed the biggest pocket volume, with 513.34 Å$^3$ and 31 residues, whereas the *L. major* pocket was the smallest (134.27 Å$^3$ and 10 residues). The *L. major* model showed a greater number of side chains facing inward to the binding site.

**Table 1.** Residues involved in the binding site of the human and *Leishmania* subtilisin S8 domain.

| Specie | Catalytic Site | Number of Residues | Volume (Å3) | Surface (Å2) | Drugscore |
|---|---|---|---|---|---|
| Human | Asp5, His36, Ser201 | 31 | 513.34 | 562.95 | 0.78 |
| *L.amazonensis* | Asp10, His43, Ser308 | 24 | 486.93 | 644.39 | 0.82 |
| *L.major* | Asp10, His43, Ser308 | 10 | 134.27 | 188.72 | 0.18 |

In order to investigate the electrostatic potential of the SUB binding site surface, MEP was carried out by the auxiliary plug-in, PyMOL APBS Tools, for the three enzymes. They each showed different profiles (data not shown), but in the case of the *L. amazonensis* SUB,

the surrounding region of the catalytic triad was predominantly positively charged whereas the most central region of the triad was a more negatively charged region. It was also possible to observe that the region near the catalytic histidine would be a less negatively charged location compared with the surface next to the other serine and aspartate of the catalytic triad.

Docking results suggest similar conformation of PF-429242 inside the binding sites of the three models (Figure 3), promoting an ionic interaction to the aspartate residue from the catalytic site. The resultant complexes also indicate that the inhibitor interacts through a hydrogen bond with the histidine residue from the catalytic site of the human and *L. major* SUBs (His36 and His43, respectively), and produces hydrophobic contacts with phenylalanine, threonine, leucine, and tyrosine residues.

PF-429242 conformation does not interact with the Asn134 in the human S1P. However, this corresponding residue in the *Leishmania* enzymes is an arginine (Arg195). This mutation to a charged residue may be a crucial difference that could be explored further in the development of new selective drugs.

### 3.5. Effect of the Inhibitor PF-429242 in Mammalian Cells and L. amazonensis

As in silico analysis suggested PF-429242 binding to *Leishmania* SUB, we next evaluated the effect of this inhibitor on peritoneal macrophages and *L. amazonensis* survival through in vitro assays. PF-429242 had a significant effect against promastigote forms of *L. amazonensis* in vitro, with $IC_{50}$ values of $3.07 \pm 0.20$, $0.83 \pm 0.12$, $2.02 \pm 0.27$ and $5.83 \pm 1.2$ µM against the LTB0016, PH8, Josefa, and LV78 strains, respectively (Table 2). On the other hand, PF-429242 had low cytotoxic effects on mammalian cells, peritoneal macrophages from BALB/c mice, with a $CC_{50}$ value of 170.30 µM. Significant toxicity was only seen for a concentration of 200 µM (Figure 4). From these values, the selectivity index (SI) towards the promastigotes over the host cells could be calculated (SI = 55.47, 205.18, 84.31 and 29.15 for LTB0016, PH8, Josefa, and LV78, respectively) (Table 2). In contrast, PF-429242 showed low activity against *L. amazonensis* intracellular amastigotes, with an $IC_{50}$ >100 µM against LTB0016, PH8, and Josefa strains, and against axenic amastigotes of the LV78 strain at $94.12 \pm 2.8$ µM (Table 2).

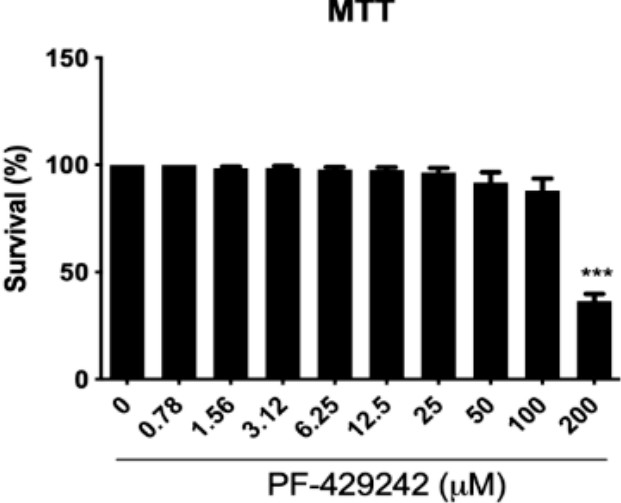

**Figure 4.** Cytotoxicity of PF-429242 in mammalian cells. Peritoneal macrophages from BALB/c mice were treated with different concentrations of PF-429242 for 72 h at 37 °C. Cytotoxicity was evaluated by MTT assays. Graphs represent the percentage of viable macrophages after treatment. Statistical analysis by one-way ANOVA followed by Tukey's post-test was performed. Experiments were performed in triplicate and graphs are representative of three independent experiments. *** $p < 0.001$.

**Table 2.** Effect of PF-429242 against peritoneal macrophages and promastigotes and intracellular amastigotes of *L. amazonensis*.

| | Cytotoxicity against Peritoneal Macrophages—CC$_{50}$ (μM) | Antileishmanial Activity against | | | | | | | | | | | | | | | | |
|---|---|---|---|---|---|---|---|---|---|---|---|---|---|---|---|---|---|---|
| | | *L. amazonensis* | | | | | | | | | | | | | | | | |
| | | Promastigotes | | | | | | | | Intracellular Amastigotes | | | | | | | | Amastigote-like |
| | | IC$_{50}$ (μM) | | | | SI | | | | IC$_{50}$ (μM) | | | | SI | | | | IC$_{50}$ (μM) |
| | | LTB0016 | PH8 | JOSEFA | LV78 | LTB0016 | PH8 | JOSEFA | LV78 | LTB0016 | PH8 | JOSEFA | LV78 | LTB0016 | PH8 | JOSEFA | LV78 | LV78 |
| PF-429242 | 170.30 ± 6.41 | 3.07 ± 0.20 | 0.83 ± 0.12 | 2.02 ± 0.27 | 5.83 ± 1.2 | 55.47 | 205.18 | 84.31 | 29.15 | >100 | >100 | >100 | - | - | - | - | - | 94.12 ± 2.84 |

Furthermore, PF-429242 significantly altered the survival of the promastigotes in concentrations greater than 1 μM (Figure 5A,B). However, against intracellular amastigotes, this compound only significantly reduced the survival at 100 μM for all three strains, as well as at 10 μM for the PH8 strain (Figure 5C). The same was seen for the axenic amastigotes of the LV78 strain (Figure 5D).

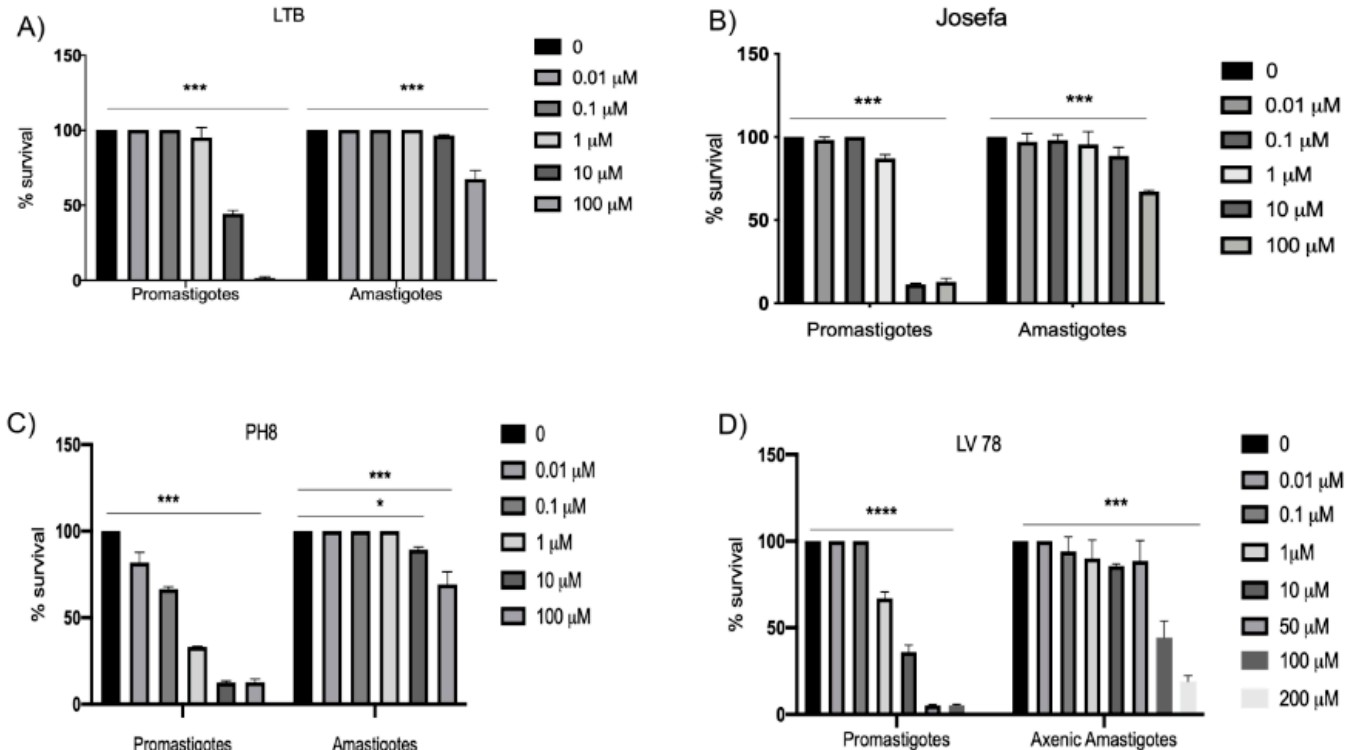

**Figure 5.** Survival percentage of promastigotes and intracellular amastigotes treated with PF-429242. *L. amazonensis* promastigotes, and intracellular or axenic amastigotes were treated with different concentrations of PF-429242 for 72 h. (**A**) LTB, (**B**) Josefa, (**C**) PH8, and (**D**) LV78. Promastigotes measured by resazurin or MTT. Infected macrophages were fixed, stained with Panotico and counted on a microscope. The total number of amastigotes within 100 macrophages were assessed. Graphs show the average from three independent experiments. Statistical analysis by one-way ANOVA followed by Tukey's post-test was performed to compare results with the untreated control. **** $p < 0.0001$, *** $p < 0.001$, * $p < 0.05$.

### 3.6. Expression of Subtilisin in Life Cycle Forms

Although PF-429242 had the capacity to impair the growth of promastigotes, as shown in Figure 5, this inhibitor was less effective against the amastigote form (Figure 5D). Thus, we evaluated the expression of SUB in both the promastigote and amastigote forms (Figure 6) by flow cytometry under permeabilized conditions. Greater expression of this enzyme was detected in the amastigote than in promastigote ($16.73 \pm 0.65$ and $2.98 \pm 0.30$, $p < 0.0035$).

Although PF-429242 was not effective against the amastigote form of *L.amazonensis*, it did have an effect against the promastigotes. This could be due to the fact that the amastigotes appear to possess more of the target, SUB. This inhibitor could be used as the basis for developing more effective antileishmanial drugs.

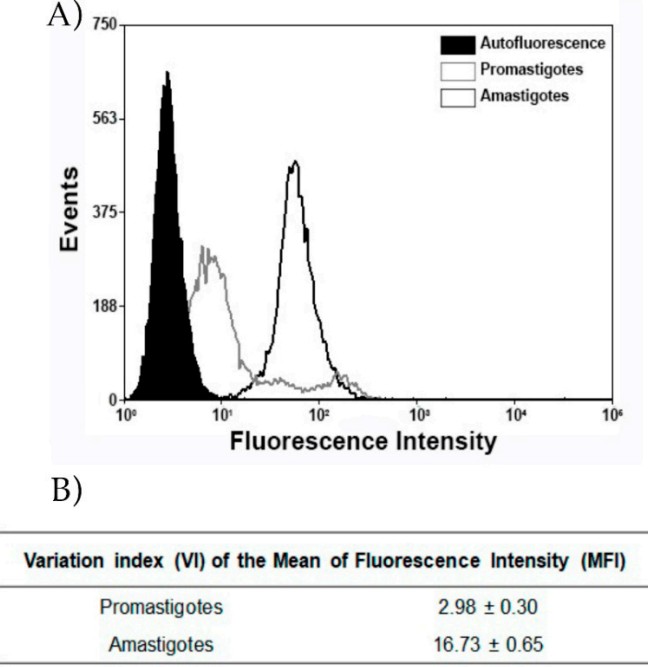

**Figure 6.** Intracellular protein expression of subtilisin in promastigotes and axenic amastigotes. (**A**) Flow cytometry analysis of *L. amazonensis* axenic promastigotes and amastigotes (RAT/BA/74/LV78). Cells were labeled with the anti-subtilisin antibody (1:400) and an Alexa 488 secondary antibody (green) (1:750). (**B**) The variation index of the mean fluorescent intensity (MFI) was obtained by the division of the MFI from labeled parasites by the non-stained autofluorescence control (black fill).

## 4. Discussion

The role of peptidases has been studied in various pathological contexts, including infection by parasitic helminths [42] and fungi [43], and cancer [44]. In protozoans, peptidases have been reported to have several functions, such as evasion of the immune system, mediating interaction with host cells, and ensuring their infectivity, virulence and nutrition [45], as in the case of the etiologic agents of malaria and toxoplasmosis. The serine protease subtilisin is required for many stages of the parasite life cycle.

In *Plasmodium*, the importance of subtilisin-like protease 1 (SUB) is related to the continuation of the life cycle for several species of the genus [46]. When using a potent inhibitor of the SUB propeptide-like protein, the egress of the merozoite form of *Plasmodium* was impaired, thereby affecting its survival [47].

The subtilisin-like peptidase is the most well-known subtilase found in *Plasmodium*, mainly in the early stages in exomes, merozoite-secreting organelles [48]. In *Toxoplasma gondii*, TgSUB1 cleaves several microneme protein complexes on the surface of the parasite and is responsible for activating gliding motility and host invasion [49].

A study in 2010 revealed the importance of SUB for different species of *Leishmania* [22]. In this study, it was observed that the subtilisin-like protease gene could only be knocked out in *L. donovani* but not in *L. major*, in which only one allele could be silenced, suggesting that it may be essential for the survival of this species [22]. Furthermore, these *Leishmania* mutants demonstrated greater sensitivity to hydroperoxides [22], as well as reduced viability during in vivo infection, for both the heterozygotic *L. major* line and the SUB-deficient *L. donovani*. Thus, it can be said that this serine protease plays a role in *Leishmania* virulence. Additionally, subtilisin is involved in promastigote to amastigote differentiation that directly affects the virulence of the parasite [22].

However, despite the observed importance of this protease in *Leishmania* parasites, the cell localization had, until now, not been determined. In our study, we identified the presence of SUB in promastigotes of both *L. amazonensis* and *L. major*, using an antibody produced against the highly conserved catalytic domain, with a particularly high concen-

tration of this protease in the region of the flagellar pocket. The flagellar pocket serves several functions for *Leishmania*, including motility for migration, attachment to the midgut microvilli and stomodeal valve of the vector, and potential sensory functions [50]. The process of capturing macromolecules from the parasitophorous vacuole of the host cells by amastigotes is mediated by the flagellar pocket [51]. According to the location of subtilisin we hypothesise that this enzyme may participate in a function in the flagellar pocket, perhaps related to parasite nutrition or interaction with the host; however, this must be elucidated. Previously, studies have reported that targeting proteins in the flagellum can damage the flagellar pocket of *L. mexicana*, thus impairing the viability of the parasites in the host macrophage [52].

The protein sequence of subtilisin for the *L. amazonensis* reference strain M2269 is 1467 amino acids, which equates to approximately 154 kDa. Using antibodies against the SUB catalytic domain, three bands were identified in the lysates from *L. amazonensis* LTB0016. The largest band (144 kDa) was approximately the size of that expected for the reference strain. The two smaller bands (129 kDa and 57 kDa) could indicate cleavage of the full-length protein. A cleavage site to remove a signal peptide has been predicted for subtilisin of *L. donovani* [22]. Although other potential cleavage sites are not yet known.

Studies investigating the serine proteases of *L.amazonensis* have identified enzymes around the same size as the smallest band detected in the present study, although these serine proteases have not been classified nor have the functions been investigated. Using gel zymography, a serine protease of 68 kDa was identified in the PH8 strain *L.amazonensis* [53], whereas a 56 kDa serine protease was identified in the same strain in a different study [54]. Similar to subtilisin, this 56 kDa serine protease was detected in the flagellar pocket [54]. This serine protease was also found in cytoplasmic vesicles and released into the extracellular environment through the flagellar pocket.

Since the proper functioning of the flagellar pocket is vital to *Leishmania* spp., targeting a specific protein known to localize in this region, that may have a role in the flagellar pocket, could be one route to consider in terms of drug development. Silva-Lopez et al. (2007) demonstrated that generic serine protease inhibitors (Bza, TPCK, and TLCK) impaired the viability and altered the morphology of *L. amazonensis* promastigotes [50]. Ultrastructural changes have been observed in *Leishmania* upon treatment with Bza and TPCK in particular, with alterations in the flagellar pocket region and membrane, including bleb formation. However, the effects caused by TPCK were more pronounced than those of Bza, with the detection of intracellular vesicular bodies [50]. Ultrastructural changes have been observed in *Leishmania* upon treatment with Bza and TPCK in particular, with alterations in the flagellar pocket region and membrane, including bleb formation. However, the effects caused by TPCK were more pronounced than those of Bza, with the detection of intracellular vesicular bodies [50].

In a recent study by our group, we evaluated the effect of PF-429242 during in vitro infection with *L. infantum* [29]. In this work, we demonstrated that PF-429242 acts on *L. infantum* promastigotes and intracellular amastigotes inducing morphological changes in the flagellum, which corresponds with our data of the cellular localization of subtilisin [29].

In the current study, we used PF-429242 in molecular docking studies with the binding site of the *L. major* and *L. amazonensis* SUB as well as the human S1P (or MBTP1). The catalytic domain of the *L. amazonensis* SUB shared 27.47% identity with the human MBTP1 (Table S1). Based on the similarity observed with S1P, the amino acids that potentially make up the catalytic triad have been highlighted in the sequences given, as seen in Figure S1.

The results indicate that PF-429242 adopts a similar conformation with the in silico modeled active sites of all three enzymes. The catalytic aspartate residue is key to the binding, since it promotes an ionic interaction to the ligand which stabilizes the complex. However, Asn134 in S1P is not conserved in the *Leishmania* SUBs, as it is replaced by Arg195. One aspect to consider for an antileishmanial drug is one which would inhibit the parasite protein, but not affect the human homologue. Thus, since Arg and Asn residues

have different physical-chemical properties, this could be explored further to optimize the activity and selectivity of new inhibitors.

PF-429242 showed low toxicity to murine macrophages but was active against the promastigote form of four different strains of *L. amazonensis*. PF-429242 was demonstrated to be less active against the intracellular amastigote compared with the extracellular forms. This can be explained by the fact that the activity of serine protease in *L. amazonensis* is greater in promastigotes than in amastigotes [51]. Moreover, the drug is very hydrophilic, which can be a factor that makes it difficult for it to reach the amastigote. In addition, PF-429242 has already been shown to have a high clearance rate (75 mL/min/kg) and thus was administered in the animal models of one study every 6 h for 24 h [41]. According to our data, *L. amazonensis* amastigotes also have a higher amount of SUB when compared with promastigotes, which may therefore require a higher concentration of the compound for enzymatic inhibition.

Depending on the *Leishmania* species, amastigote forms can be hosted individually in small, tight parasitophorous vacuoles (PVs) or grouped into large, loose PVs, which may explain the need to test higher drug concentrations or improved delivery across membranes [55]. Furthermore, the need to cross the host cell membrane and the parasite membrane can result in loss of the compound before reaching the amastigotes [56].

Recently, PF-429242 treatment of *L. infantum*-infected macrophages was effective [29] but this was not observed for the *L. amazonensis*-infected macrophages. This is likely due to the large parasitophorous vacuoles present in *L. amazonensis*-infected macrophages that generates a resistant mechanism in tryparedoxin peroxidase isoform, which is different from parasitophorous vacuoles of *L. infantum* [57].

Drug encapsulation strategies could be the focus of future studies to improve the delivery of this inhibitor for the treatment of infections caused by the different *Leishmania* species. In fact, PF-429242 has already been encapsulated in a lymphocytic choriomeningitis virus (LCMV) in an attempt to enhance the efficiency of drug delivery and its ability to eliminate chronic infections, wherein the authors obtained satisfactory results [26].

Here, we show for the first time the cellular location of the subtilisin-like serine protease in the flagellar pocket of *L.amazonensis* and that the serine protease inhibitor, PF-429242, can affect promastigote survival but not that of the amastigotes.

## 5. Conclusions

Here, we show for the first time the cellular location of the subtilisin-like serine protease in the flagellar pocket of *L. amazonensis* and that the serine protease inhibitor, PF-429242, can affect promastigote survival but not the amastigotes. However, these findings will allow the creation of strategies to increase drug delivery to maintain efficacy against the *L. amazonensis* intracellular forms and during in vivo infection.

**Supplementary Materials:** The following supporting information can be downloaded at: https://www.mdpi.com/article/10.3390/cimb44050141/s1, Table S1: Percentage of similarity between the catalytic domain sequences. The similarity was obtained by Clustal O (1.2.4); Figure S1: Multiple sequence alignment of trypanosomatid and human subtilisins (MBTP1). Alignment of the catalytic domain sequences of SUBs from *L. amazonensis*, *L. braziliensis*, *L. donovani*, *L. infantum*, *L. major*, *L. mexicana*, *L. panamensis*, *T. cruzi*, *T. brucei* and *Homo sapiens*. Sequences were obtained from the UnitProt database, and the alignment was performed using the Clustal O (1.2.4) program. * indicates amino acids are equal in the same position: identify high similarities, moderate similarities and yellow highlighting indicates the amino acids that potentially form the catalytic triad; Figure S2: Cloning, expression and purification of the catalytic domain of *L. major* subtilisin. (A) PCR amplification of LMJF_13_1040; lane 1, the complete ORF and lane 2, the catalytic domain only. (B) Purification of the recombinant SUB catalytic domain expressed by *E. coli* detected on a 12% SDS-PAGE gel. (C) Detection of the recombinant *L. major* SUB catalytic domain with the anti-subtilisin antibody. Lane 1, anti-subtilisin immunized serum and lane 2, non-immunized serum; Figure S3: Immunolocalization of subtilisin in *L. major* by confocal microscopy. (A) *L. major* promastigotes were fixed, incubated with anti-subtilisin antibody (1:100) and then incubated with Texas Red-labeled secondary antibody

(1:1000). Images were acquired by excitation at 590 nm and emission at 615 nm. The nuclei were stained blue with DAPI. (B) Differential interference contrast (DIC) image of figure (A); Figure S4: Two-dimensional representation of the PF-429242 structure generated in PubChem Sketcher V2.4.

**Author Contributions:** Conception and idea: H.L.d.M.G., Design of the study: P.S.G., M.P.D.C., P.d.A.M., S.G.D.-S. and H.L.d.M.G. Acquisition of data: P.S.G., M.P.D.C., P.d.A.M., J.V.M.P.d.S., A.M.d.F.-M., V.E.-V., V.V.d.A.-N. and H.L.d.M.G. Analysis and interpretation of data: P.S.G., M.P.D.C., V.E.-V., P.d.A.M., A.G., V.V.d.A.-N., A.P.C.d.A.L. and H.L.d.M.G. Support and material: V.V.d.A.-N., V.E.-V., D.C.O.G., S.G.D.-S., A.P.C.d.A.L. and H.L.d.M.G. Drafting the article: P.S.G., M.P.D.C., P.d.A.M., A.G., A.P.C.d.A.L. and H.L.d.M.G. Revising the article critically for important intellectual content: P.S.G., P.d.A.M., D.C.O.G., A.C.R.S., S.G.D.-S., A.P.C.d.A.L. and H.L.d.M.G. All authors have read and agreed to the published version of the manuscript.

**Funding:** This research was funded by Fundação Carlos Chagas Filho de Amparo à Pesquisa do Estado do Rio de Janeiro (FAPERJ) grant number E-26/202.421/2017 and E-26/202.942/2019. Conselho Nacional de Desenvolvimento Científico e Tecnológico. Universidade Federal do Rio de Janeiro by grant PIBIC/PIBITI.

**Institutional Review Board Statement:** The study was conducted according to the guidelines of the Federal University of Rio de Janeiro and approved by Ethics Committee on Animal Experimentation (CEUA 080/2018).

**Informed Consent Statement:** Not applicable.

**Data Availability Statement:** Not applicable.

**Acknowledgments:** This work was supported by grants from Conselho Nacional de Desenvolvimento Científico e Tecnológico do Brasil (CNPq), Fundação de Amparo à Pesquisa do Estado do Rio de Janeiro (FAPERJ), Universidade Federal do Rio de Janeiro, and Fiocruz. We thank Bartira Rossi Bergmann, Suzana Côrte-Real, Dario Kalume and Carlos Rangel Rodrigues for providing support and supplies.

**Conflicts of Interest:** The authors declare no conflict of interest. The authors declare that the research was conducted in the absence of any commercial or financial relationships that could be construed as a potential conflict of interest.

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
