# Peer review of "Subtilisin of Leishmania amazonensis as Potential Druggable Target: Subcellular Localization, In Vitro Leishmanicidal Activity and Molecular Docking of PF-429242, a Subtilisin Inhibitor"

_cimb, doi:10.3390/cimb44050141_

Round 1

Author Response

Dear, reviewer

Best regards

Reviewer 2 Report

This manuscript proved the inhibitor PF-429242 of serine proteases significantly inhibited the growth of L. amazonensis promastigotes of four different strains, in addition, the authors used different methods to illustrate the mechanism behind the effect of this inhibitor. This work proved this drug might serve as a tool to explore pharmacological and potentially leishmanicidal targets. However, there are some limitations that the authors must solve before the manuscript might be accepted for publication.

My detailed comments are as follows:

  1. Although serine proteases have a canonical catalytic triad, which is composed of three highly conserved residues histidine, serine, and aspartic in the active site, some variations were discovered, e.g. Ser-His-His catalytic triad in a serine protease of cytomegalovirus (Nature. 1996;383:275-279, FASEB Journal. 2021;35:e21259.).
  2. The authors determined the effect of the inhibitor PF-429242 against promastigote forms of L. amazonensis in vitro. I wonder why did not determine the inhibitory effect of PF-429242 against subtilisin.
  3. The authors used the Robetta server to predict the 3D structure of proteins and used the DogSiteScorer server to predict the pocket volume and residues involved in the active binding site. If any experimental structures (X-Ray Crystallography) were determined?

Minor error:

  1. Page 5, line 239, there are two different typefaces.
  2. The molecular weight of the standard proteins should be labeled in Figure 1.

Author Response

Dear, reviewer

Best regards

Round 2

Reviewer 1 Report

the authors addressed the issues found during the reivison.